analytical chemistry

ET-26-HCl, metabolites, plasma protein binding, pharmacokinetics, tissue distribution, excretion

**Author for correspondence:**
Ling Wang
e-mail: rebeccawang312@gmail.com

This article has been edited by the Royal Society of Chemistry, including the commissioning, peer review process and editorial aspects up to the point of acceptance.

# Metabolite identification, tissue distribution, excretion and preclinical pharmacokinetic studies of ET-26-HCl, a new analogue of etomidate

Lu Yu[1], Xu Chen[1,2], Wen Sheng Zhang[3], Liang Zheng[1], Wen Wen Xu[1], Ming Yu Xu[1], Xue Hua Jiang[1] and Ling Wang[1]

[1]Key Laboratory of Drug-Targeting and Drug Delivery System of the Education Ministry, Sichuan Engineering Laboratory for Plant-Sourced Drug and Sichuan Research Center for Drug Precision Industrial Technology, West China School of Pharmacy Sichuan University, Chengdu 610064, People's Republic of China
[2]Chengdu Women and Children Central Hospital, Chengdu, Sichuan 610041, People's Republic of China
[3]Anesthesia and Critical Aid Laboratory, Conversion Neuroscience Center, West China Hospital of Sichuan University, Chengdu, Sichuan 610041, People's Republic of China

LW, 0000-0003-3220-6065

ET-26-HCl, a novel anaesthetic agent with promising pharmacological properties, lacks extensive studies on pharmacokinetics and disposition *in vitro* and *in vivo*. In this study, we investigated the metabolic stability, metabolite production and plasma protein binding (PPB) of ET-26-HCl along with its tissue distribution, excretion and pharmacokinetics in animals after intravenous administration. Ultra-high performance liquid chromatography–tandem quadrupole time-of-flight mass spectrometry identified a total of eight new metabolites after ET-26-HCl biotransformation in liver microsomes from different species. A hypothetical cytochrome P450-metabolic pathway including dehydrogenation, hydroxylation and demethylation was proposed. The PPB rate was highest in mouse and lowest in human. After intravenous administration, ET-26-HCl distributed rapidly to all tissues in rats and beagle dogs, with the highest concentrations in fat and liver. High concentrations of ET-26-acid, a major hydroxylation metabolite of ET-26-HCl, were found in liver, plasma and kidney. Almost complete clearance of ET-26-HCl from plasma

occurred within 4 h after administration. Only a small fraction of the parent compound and its acid form were excreted via the urine and faeces. Taken together, the results added to a better understanding of the metabolic and pharmacokinetic properties of ET-26-HCl, which may contribute to the further development of this drug.

# 1. Introduction

Etomidate, R-1-(1-ethylphenyl) imidazole-5-ethyl ester, is an imidazole-based anaesthetic agent (figure 1a), commonly used as a short-acting intravenous agent for anaesthesia and sedation because of its favourable myocardial performance [1,2]. However, the use of etomidate remains clinically limited and controversial due to its persistent adrenal suppression [3–5].

ET-26-HCl, (R)-2-methoxyethyl1-(1-phenylethyl)-1H-imidazole-5-carboxylate hydrochloride, an analogue of etomidate, is a novel anaesthetic agent synthesized by the Neuroscience Research Center Anesthesia and Critical Emergency Research office at West China Hospital of Sichuan University (figure 1b). Compared to etomidate, ET-26-HCl exhibited superior anaesthetic property, reduced adrenal suppression and optimal myocardial performance [6–10]. Given these outstanding features, determining the pharmacokinetic and *in vivo* disposition of ET-HCl-26 becomes imperative. ET-26-acid (etomidate acid, figure 1c) is reported as the main inactive ester hydrolysis-generated metabolite of ET-26-HCl [7]. However, biotransformation of ET-26-HCl in other metabolic pathways, especially the cytochrome P450 (CYP) system remains understudied. It was reported that ET-26-HCl was rapidly cleared within 4 h of intravenous administration in rats, and the metabolite, ET-26-acid, had a higher plasma exposure than the parent drug [11].

We conducted a comprehensive disposition study and preclinical pharmacokinetics (PK) to support the further development of this new drug. In this study, we evaluated the metabolic stability of ET-26-HCl in liver microsomes from different species and identified CYP metabolites using ultra-high performance liquid chromatography–tandem quadrupole time-of-flight mass spectrometry (UPLC-QTOF-MS). PK of ET-26-HCl was conducted in beagle dogs following intravenous administration. In addition, we investigated biodistribution and excretion profile of ET-26-HCl and ET-26-acid and plasma protein binding (PPB) *in vitro*.

# 2. Material and methods

## 2.1. Materials

ET-26-HCl and its main metabolite ET-26-acid (purity greater than 99.0% by HPLC) were provided by the Anesthesia and Critical Aid Laboratory, Conversion Neuroscience Center, West China Hospital of Sichuan University. Internal standard (IS), gabapentin (GBP, figure 1d) was obtained from Palm Medicine Chemicals Ltd (Nanyang, Zhejiang, China). HPLC grade methanol was purchased from Sigma-Aldrich (Merck KGaA, Darmstadt, Germany). Formic acid was purchased from Dikma Technologies, Inc. (Lake Forest, CA, USA). Liver microsomes of human, monkey, dog, rat and mouse were purchased from Becton, Dickinson and Company (Franklin Lake, New Jersey, USA).

## 2.2. ET-26-HCl and ET-26-acid analysis in biological samples

Concentrations of ET-26-HCl and ET-26-acid in biological samples were determined by liquid chromatography–tandem mass spectrometry (LC–MS/MS) after deproteinization with methanol. The bioanalytical method for simultaneous quantification of ET-26-HCl and ET-26-acid in the different matrix was fully validated according to FDA guidelines (electronic supplementary material, figure S1 and tables S1 and S2) in our previous study [11,12]. Briefly, a 100 µl sample (plasma, tissue homogenate, bile, urine and faeces suspension) were spiked with 20 µl of IS solution and vortexed for 30 s. The mixture was then processed for protein precipitation with 300 µl methanol, followed by centrifugation at 12 000 r.p.m. for 5 min. Supernatant was collected and analysed by LC–MS/MS system.

The LC–MS/MS was performed as described previously [11]. Briefly, ET-26-HCl, ET-26-acid and GBP were separated using a CAPCELL PAK $C_{18}$ column (50 × 2.00 mm, 5.00 µm) as well as a Security Guard Cartridge ($C_{18}$, 4 mm × 3.00 mm i.d., Phenomenex). The mobile phase consisting of 0.3% formic acid

**Figure 1.** Chemical structures of (*a*) etomidate, (*b*) ET-26-HCl, (*c*) ET-26-acid and (*d*) gabapentin.

water (A), and methanol (B) was pumped at a flow rate of 0.6 ml min$^{-1}$. The gradient elution was achieved according to the following protocol: 0 min to about 0.50 min, A/B (41/9, v/v); 0.51 min to about 1.5 min, A/B (1/9, v/v); and 1.51 min to about 2.00 min, A/B (41/9%, v/v). Mass spectrometry was conducted on an API 3000 mass spectrometer (Sciex, Framingham, MA, USA) equipped with an electrospray ionization (ESI) source operating in positive mode. The MS/MS system was in the multiple reaction monitoring (MRM) mode, using the *m/z* transitions of 275.6/170.9 for ET-26-HCl, 217.7/113.1 for ET-26-acid and 172.5/154.3 for GBP (IS), respectively. The API 3000 mass parameters were set according to the following protocol: curtain gas, 6 psi; collision gas, 8 psi; nebulizer, 4 psi; ion spray voltage, 3000 V and ion source temperature, 450°C. The autosampler was conditioned at 4°C, and the injection volume was 5 µl. The linear ranges of this analytical method for ET-26-HCl and for ET-26-acid from all biological samples are shown in electronic supplementary material, table S3.

## 2.3. Metabolic stability in liver microsomes

*In vitro* metabolic stability of ET-26-HCl was estimated in liver microsomes from human, monkey, dog, rat and mouse. A mixture of 100 mM ET-26-HCl, 2.0 M VIVID, 5.0 M MgCl$_2$ and liver microsomes (human, monkey, dog, rat and mouse, 0.33 mg microsomes protein/ml) in Tris-HCl phosphate buffer (pH 7.4) was pre-incubated for 10 min at 37°C before the addition of 1.0 mM NADPH and then quenched with 450 µl methanol at 0, 7, 17, 30 and 60 min. Samples were analysed by LC–MS/MS as mentioned above.

$$\text{CL}_{\text{int, in vitro}} = \frac{0.693/t_{1/2}}{\text{C}_{\text{protein}}}, \tag{2.1}$$

$$\text{CL}_{\text{int, in vivo}} = \frac{\text{CL}_{\text{int, in vitro}} \times \text{Houston} \times \text{LW}}{1000} \tag{2.2}$$

and

$$\text{CL}_{\text{hep in vivo}} = \frac{\text{HBF} \times \text{fu} \times \text{CL}_{\text{int, in vivo}}}{\text{HBF} + \text{fu} \times \text{CL}_{\text{int, in vivo}}}. \tag{2.3}$$

Where C$_{\text{protein}}$ is microsomal protein concentration (mg ml$^{-1}$); Houston is Houston factor (45 mg of microsomal protein g$^{-1}$ liver); LW is liver weight (g) (per species); HBF is hepatic blood flow (ml min$^{-1}$) (per species); fu (two places) and later is unbound fraction (fu = 1 − %protein binding).

## 2.4. Identification of major metabolites in liver microsomes

The major metabolites of ET-26-HCl in liver microsomes from mouse, rat, beagle dog, monkey and human were determined by the UPLC-Q-TOF-MS system.

Reaction mixture containing 10 µM ET-26-HCl, 5 M MgCl$_2$ and 1 mg ml$^{-1}$ liver microsomes in Tris-HCl phosphate buffer (pH 7.4) was pre-incubated for 6 min at 37°C, followed by the addition of NADPH. After incubation for 1 h, methanol was added to stop the reaction. Samples were centrifuged

at 11 000 r.p.m. for 5 min; supernatants were collected and air dried at 40°C. Residues were dissolved in 100 µl acetonitrile/water (1:1, v/v) and analysed by the UPLC-Q-TOF-MS.

The UPLC-Q-OF-MS system was equipped with a Waters Synapt G2-Si quadrupole time-of-flight mass spectrometer (Q-TOF-MS), an ESI source and an ACQUITY UPLC H-CLASS liquid chromatography system. The chromatographic separations were achieved on an ACQUITY UPLC BEH C$_{18}$ column (2.1 × 100 mm, i.d. 1.7 µm, Waters, USA) kept at 45°C and a flow rate of 0.4 ml min$^{-1}$. The mobile phase consisted of 5 mM ammonium acetate with 0.1% formic acid (A) and acetonitrile with 0.1% formic acid (B). The optimized gradient elution programme followed was 0 min to 2.1 min, A/B (19/1, v/v); 2.1 min to 7.3 min, A/B (1/4, v/v); 7.3 min to 8.0 min, A/B (1/9, v/v); and 8.0 min to 10 min, A/B (19/1, v/v). The injection volume was 5 µl. The mass experiment was operated in positive ion modes under the following parameters: turbo spray temperature, 120°C; ion spray voltage, 3.0 kV. Nitrogen was used as the nebulizer and the auxiliary gas; meanwhile, the nebulizer gas was set to 650 l h$^{-1}$ and kept at 350°C. The collision energy was set at 5–10 V and the collision energy spread was 4–6 V. TOF-MS was scanned with the mass range of $m/z$ 50–600 with 200 ms accumulation time. A 400 ng ml$^{-1}$ leu-enkephalin ($m/z$ 556.2771) was selected as an external standard for mass–charge ratio correction at a flow rate of 5 µl min$^{-1}$.

## 2.5. Plasma protein binding of ET-26-HCl

The PPB of ET-26-HCl in mouse, rat, beagle dog, monkey and human plasma was determined according to the equilibrium dialysis method [13,14]. In brief, 1 ml of control plasma samples in dialysis bag was incubated in 10 ml of PBS containing ET-26-HCl at a final concentration of $2.21 \times 10^2$, $1.77 \times 10^3$ and $3.54 \times 10^3$ ng l$^{-1}$ at 4°C with an equilibrium time of 9 h to reach the mass balance binding condition. A 100 µl aliquots of the PBS buffer and the plasma in the dialysis bag were transferred to 2 ml EP tubes and diluted to 995 µl with PBS and analysed on the LC–MS/MS system. The PPB ($f_{ppb}$) was calculated as follows [15]:

$$f_{ppb} = \frac{D_{in} - D_{out}}{D_{in}} \times 100\% \,,$$

where $D_{in}$ is the plasma drug concentration in the dialysis bag and $D_{out}$ is the dialysate drug concentration.

## 2.6. Pharmacokinetics of ET-26-HCl in beagle dogs

Beagle dogs, weighing 9–11 kg, were purchased from Sichuan Industrial of Antibiotics, China National Pharmaceutical Group Corporation (Chengdu, China). A 3 × 3 crossover experimental design with 24 beagles (equal number of male and female dogs) randomly divided into six groups was conducted. Beagle dogs in the six groups received three doses of ET-26-HCl: 1.045 mg kg$^{-1}$, 2.090 mg kg$^{-1}$ and 4.180 mg kg$^{-1}$ as low (L), medium (M) and high (H) dose. Blood samples (2 ml) were collected from the elbow vein in 4 ml heparinized tubes before dosing, and at 1, 3, 5, 10, 20, 30, 120 and 180 min post-dosing. All blood samples were centrifuged at 8000 r.p.m. for 10 min and plasma collected and stored at −80°C until analysis.

## 2.7. Tissue distribution of ET-26-HCl in rats

Sprague Dawley (SD) rats (200–240 g) were provided by the Animal Facility at Sichuan University (Chengdu, China). All rats received a single 4.2 mg kg$^{-1}$ dose of ET-26-HCl intravenously. Rats were sacrificed and tissues, including heart, spleen, lung, intestine, colon, kidney, brain, fat, stomach, liver, muscle, testis and ovary were collected at 0, 1, 3, 6 and 10 h post-dosing (six rats/time-point). Tissue samples were rinsed thrice in ice cold saline, and blotted dry with filter papers. Weighed amount of tissues were homogenized in normal saline (thrice the tissue weight, w/v) and homogenates were stored at −80°C until analysis.

## 2.8. Excretion of ET-26-HCl in rats

SD rats (4 male and 4 female) received 4.2 mg kg$^{-1}$ of ET-26-HCl intravenously and were housed in metabolic cages. Urine and faeces from the treated rats were collected between 0–6, 6–12, 12–24, 24–30, 30–36 and 36–48 h post-dosing. Urine volume and faeces weight were measured and recorded.

Bile fistulas of SD rats (four male and four female) were cannulated with polyethylene tube (Instech Laboratories, Inc., USA) and bile was collected at the following time-points: 0–3, 3–6, 6–9, 9–12 and 12–24 h post-dosing. The volume of the collected bile was measured and stored at −80°C until analysis.

## 2.9. Statistical analysis

### 2.9.1. Metabolism

The data were assessed using Masslynx V4.1 (Waters Corp., Milford, MA, USA).

### 2.9.2. Pharmacokinetics, tissue distribution, excretion and plasma protein binding

All calculations were performed on Microsoft Excel 2016 (Microsoft Co., USA). The Drug and Statistics 3.2.7 (DAS 3.2.7) software package (Mathematical Pharmacology Professional Committee of China, China) was used to acquire the main pharmacokinetic parameters of ET-26-HCl. All data were shown as mean ± standard deviation (s.d.). Statistical significances were calculated by one-way analysis of variance (ANOVA).

# 3. Results and discussion

## 3.1. Stability and metabolism study of ET-26- HCl *in vitro*

### 3.1.1. Metabolic stability of ET-26-HCl

ET-26-HCl was not metabolically stable in liver microsomes from monkey, dog, rat and mouse, but was relatively stable in human liver microsomes (figure 2), as assessed by the disappearance of ET-26-HCl (table 1). After incubation under the given condition, ET-26-HCl completely disappeared within 15 min in the monkey liver microsomes, followed by rat, mouse and dog. Human liver microsomes did not wholly transform ET-26-HCl at 1 h. The $CL_{int,\,in\,vitro}$ of ET-26-HCl, was 50.0, 2216, 466, 984 and 629 ml min g$^{-1}$ protein in liver microsomes from human, monkey, dog, rat and mouse, respectively.

### 3.1.2. Identification of major metabolites of ET-26-HCL in liver microsomes

The parent compound ET-26-HCl was detected only in liver microsomes samples from human and dog. It was eluted at 5.97 min with an accurate $[M + H]^+$ ion at $m/z$ 275.1395, suggesting an elemental composition of $C_{15}H_{18}N_2O_3$ (figures 3 and 4a).

#### 3.1.2.1. Metabolite M1
Metabolite M1 eluted at 5.41 min was detected in liver microsomes from human, monkey, dog, rat and mouse with $[M + H]^+$ ions at $m/z$ 259.1075 ($C_{14}H_{14}N_2O_3$), $CH_4$ less than that of ET-26-HCl (figures 3 and 4b). The fragmentation behaviour was similar to that of ET-26-HCl with ion at $m/z$ 139.0569. As shown in figure 4b, characteristic product ion at $m/z$ 157.0613 was $CH_2$ less than the fragment ion of ET-26-HCl with ion at $m/z$ 171.0773, suggesting that the methyl linked with the side chain oxygen was demethylation. Dehydrogenation might have occurred on the left side of the imidazole ring. Therefore, M1 was speculated to be the oxygen-demethylated and desaturated metabolites of ET-26-HCl.

#### 3.1.2.2. Metabolite M2
Metabolite M2 was detected in every liver microsomal preparation. It was eluted at 5.25 min, with a protonated molecular ion at $m/z$ 261.1230, indicating a molecular composition of $C_{14}H_{16}N_2O_3$, $CH_2$ less than that of the parent compound ET-26-HCl. The characteristic ions at $m/z$ 95.0225 and 139.0494, were similar to that of ET-26-HCl. Similar to M1, the ion at $m/z$ 157.0596 was 14 Da lower than that of $m/z$ 171.0773, suggesting that M2 was an oxygen-demethylated metabolite of ET-26-HCl (figures 3 and 4c).

#### 3.1.2.3. Metabolite M3
Metabolite M3 was detected at the retention time of 6.12 min only in liver microsomes from human and dog (figure 3). The formula was conjectured as $C_{15}H_{16}N_2O_3$ and showed an $[M + H]^+$ ion at $m/z$ 273.1240,

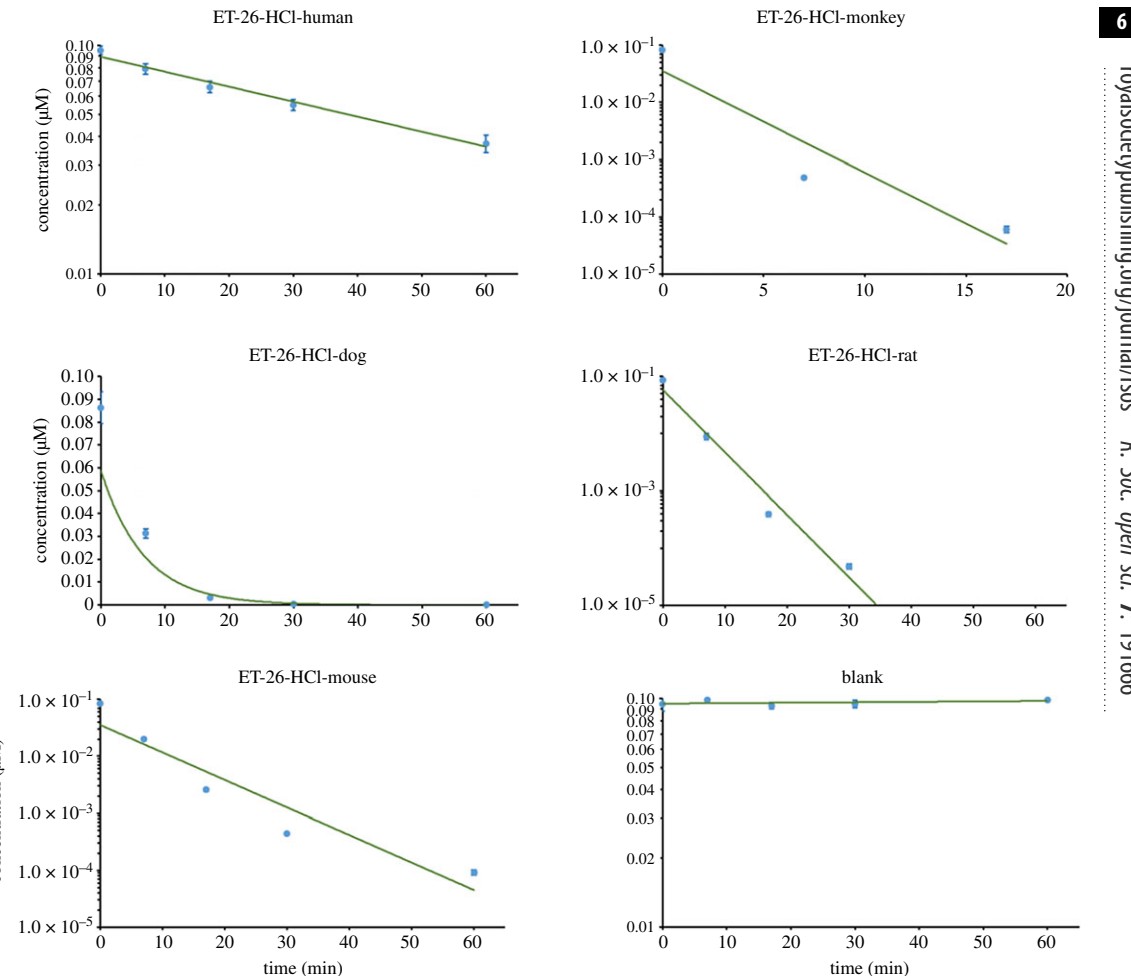

**Figure 2.** Concentration–time curves of ET-26-HCl after ET-26-HCl incubation with mice, rat, dog, monkey and human liver microsomes.

**Table 1.** Metabolic stability of ET-26-HCl in liver microsomes of different species. Data are presented as mean $\pm$ s.d., $n = 3$. $C_{protein}$, microsomal protein concentration; $t_{1/2}$, half-life; $CL_{int,\ in\ vitro}$, intrinsic clearance rate *in vitro*; $CL_{int,\ in\ vivo}$, intrinsic clearance rate *in vivo*; $C_{hep\ in\ vivo}$, hepatic clearance rate *in vivo*.

| | species | $C_{protein}$ (mg ml$^{-1}$) | $t_{1/2}$ (min) | $CL_{int,\ in\ vitro}$ (ml min$^{-1}$ g$^{-1}$ protein) | extrapolated $CL_{int,\ in\ vivo}$ (ml min$^{-1}$) | extrapolated $CL_{hep\ in\ vivo}$ (ml min$^{-1}$) |
|---|---|---|---|---|---|---|
| ET-26-HCl | human | 0.33 | 42.0 ± 1.7 | 50.0 ± 1.9 | 3823 ± 146.4 | 1077 ± 11.8 |
| | rat | 0.33 | 2.10 ± 0.2 | 984 ± 68.6 | 443 ± 30.9 | 19.1 ± 0.1 |
| | mouse | 0.33 | 3.30 ± 0.1 | 629 ± 25.7 | 42.4 ± 1.7 | 2.8 ± 0.0 |
| | dog | 0.33 | 4.50 ± 0.1 | 466 ± 12.3 | 6712 ± 117 | 296 ± 0.35 |
| | monkey | 0.33 | 0.90 ± 0.0 | 2216 ± 43.0 | 11 965 ± 232 | 168 ± 0.05 |

$H_2$ less than that of the parent compound. As shown in figure 4*d*, it had ions at *m/z* 95.0246, 113.0354 and 171.0773 which were similar to corresponding ions of ET-26-HCl. These results indicated that there was no change in the right imidazolium ring but dehydrogenation occurred on the left imidazolium ring. Thus, M3 was tentatively determined as a desaturated metabolite of ET-26-HCl.

### 3.1.2.4. Metabolites M4-1 and M4-2

Metabolites M4-1 and M4-2 were detected in all the liver microsomes except human (figure 3). The $[M + H]^+$ was ion *m/z* 275.1039 at the retention times of 5.14 and 5.36 min, respectively. According

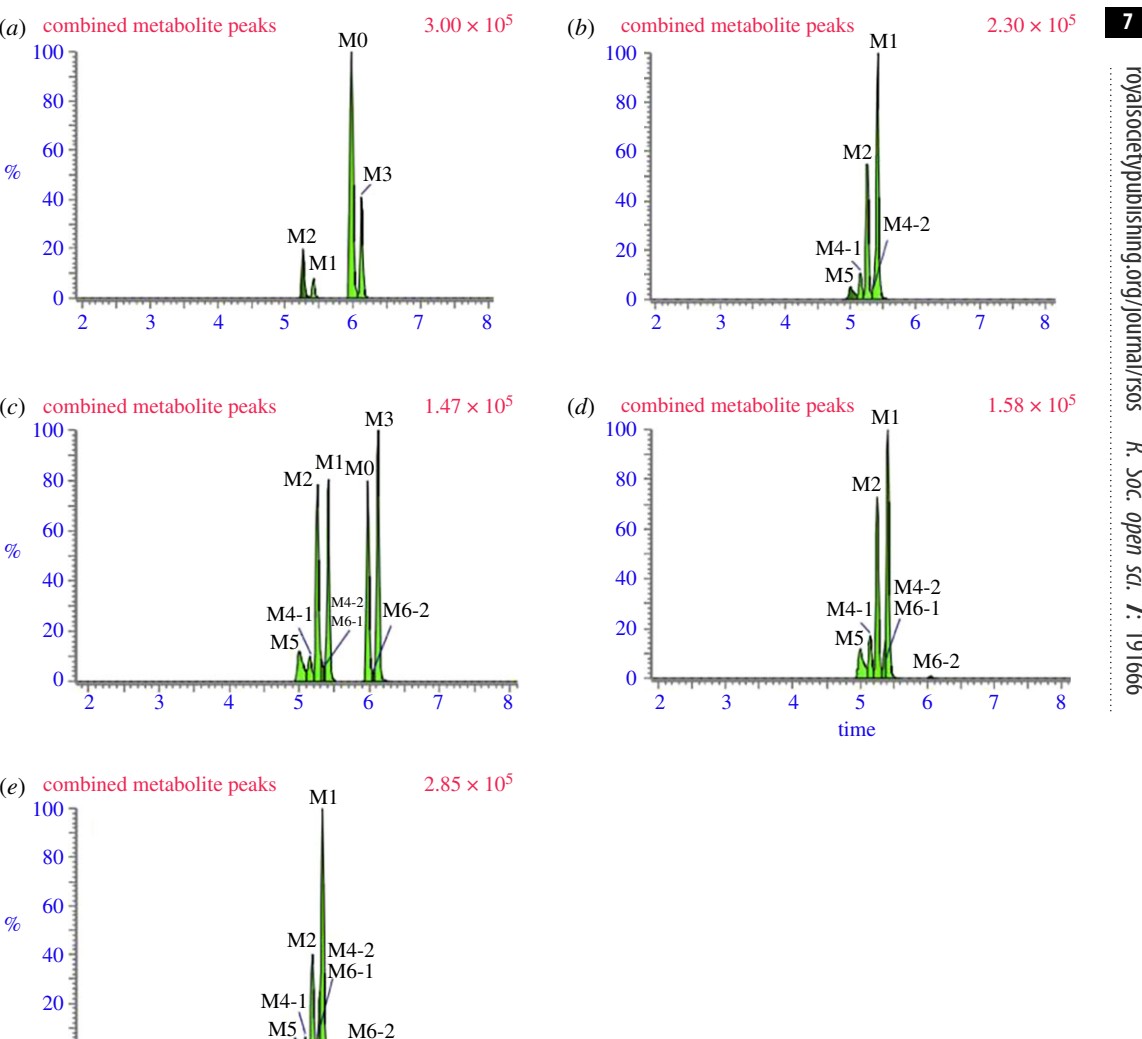

**Figure 3.** Identification of major metabolites of ET-26-HCl in liver microsomes samples from (*a*) human, (*b*) monkey, (*c*) dog, (*d*) rat and (*e*) mouse.

to the accurate mass, the molecular formulae were supposed to be $C_{14}H_{14}N_2O_4$, $CH_4$ less and O more than that of ET-26-HCl. The characteristic ions of M4-1 at *m/z* 95.0247, 113.0358 and 171.0424 were similar to ET-26-HCl, indicating that there was no change in the imidazole ring and its right side, and the deactivation of $CH_4$ and addition of O occurred on the left side of the imidazole ring. Thus, M4-1 was inferred as a demethylated, desaturated and hydroxylated metabolite of ET-26-HCl (figure 4*e*). As for M4-2, the characteristic ions at *m/z* 95.0199 was observed without ions at *m/z* 171.0773, suggesting that M4-2 was an *O*-demethylated, desaturated and hydroxylated metabolite of ET-26-HCl (figure 4*f*).

### 3.1.2.5. Metabolite M5

Metabolite M5 (*m/z* 277.1178, 5.00 min) was detected in liver microsomes from monkey, dog, rat and mouse (figure 3). The molecular formula was postulated to be $C_{14}H_{16}N_2O_4$, $CH_2$ less and O more than ET-26-HCl. The characteristic ion at *m/z* 95.0229 was similar to ET-26-HCl, which suggested no change in the imidazole ring. The fragment ion at *m/z* 173.0548, however, was O more and $CH_2$ less than the ion at *m/z* 171.0773 of ET-26-HCl, speculating demethylation and oxygenation were occurred on the right-side chain of imidazole ring. Therefore, M5 was an *O*-demethylated and hydroxylated metabolite of ET-26-HCl (figure 4*g*).

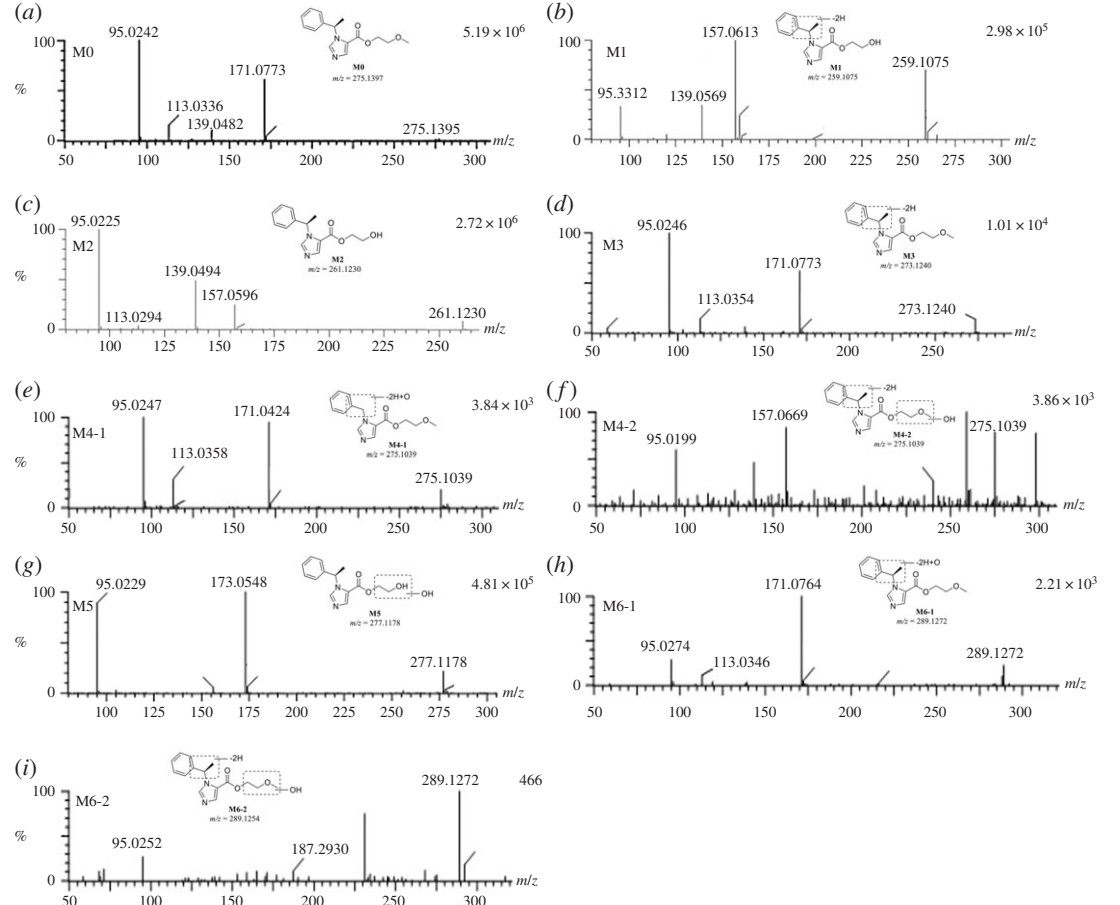

**Figure 4.** The mass spectra of (a) ET-26-HCl, (b) M1, (c) M2, (d) M3, (e) M4-1, (f) M4-2, (g) M5, (h) M6-1 and (i) M6-2.

### 3.1.2.6. Metabolites M6-1 and M6-2

Metabolites M6-1 and M6-2 were observed only in liver microsome from dog, rat and mouse (figure 3). The retention times were 5.32 and 6.06 min, respectively. The protonated molecular ion was $m/z$ 289.1272, conjectured as $C_{15}H_{16}N_2O_4$, $H_2$ less and O more than ET-26-HCl. M6-1 produced fragment ions at $m/z$ 95.0274, 113.0346 and 171.0764, similar to ET-26-HCl, which implied that the imidazole ring and its right-side chain did not change. Oxygenation and dehydrogenation occurred on the left side of the imidazole ring (figure 4h). The fragment ions at $m/z$ 95.0252 from M6-2 were similar to these of ET-26-HCl, demonstrating no changes in the imidazole ring. The fragment ion at $m/z$ 187.2930 was 16 Da higher than $m/z$ 171.0773, indicating oxygenation on the right-side chain of the imidazole ring, and thus M6-2 was a hydroxylated and desaturated metabolite of ET-26-HCl (figure 4i).

## 3.2. Determination of plasma protein binding rate

PPB rates of ET-26-HCl at 654, $2.62 \times 10^3$ and $5.23 \times 10^3$ ng ml$^{-1}$ in mouse, rat, beagle dog, monkey and human plasmas are shown in table 2. Binding rates in the plasma of mouse, rat, beagle dog, monkey and human were $94.35 \pm 1.15\%$, $83.03 \pm 1.76\%$, $66.02 \pm 4.10\%$, $62.72 \pm 3.56\%$ and $68.31 \pm 5.16\%$, respectively. A one-way single factor ANOVA and *post hoc* LSD *t*-test manifested that only the PPB rates of ET-26-HCl in mouse, rats and humans were significantly different ($p < 0.05$).

## 3.3. Pharmacokinetics in beagle dogs

Plasma concentrations of ET-26-HCl in beagles after i.v. administration of three doses are shown in figure 5, and the major PK parameters are shown in table 3. The plasma concentration ET-26-acid was almost non-detectable in the beagles. Moreover, the halflife ($t_{1/2}$) and volume of distribution ($V_d$) of ET-26-HCl increased while the clearance (CL) deceased at higher doses.

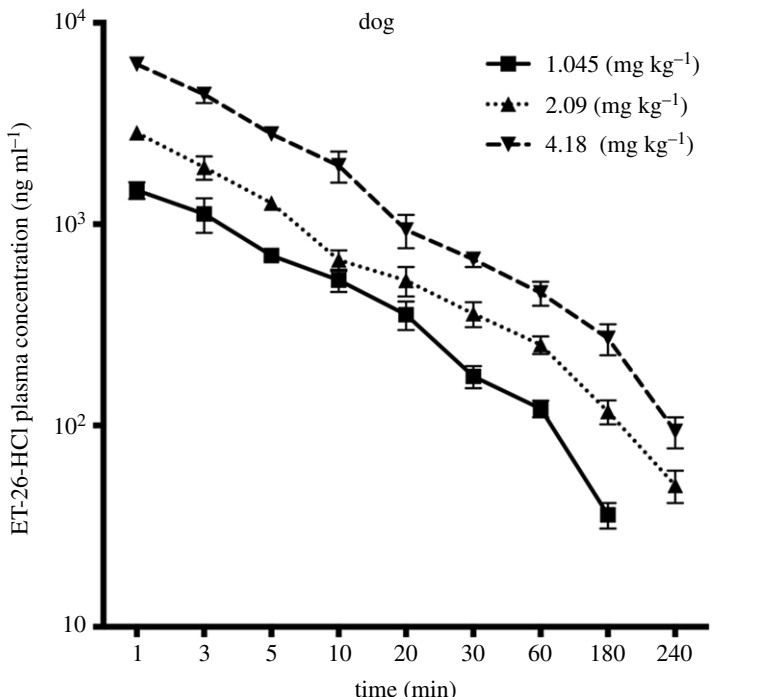

**Figure 5.** Concentration–time profiles of ET-26-HCl in beagle plasma following i.v. injection of 1.045, 2.09 and 4.18 mg kg$^{-1}$ (mean ± s.d., $n = 4$).

**Table 2.** Plasma protein binding rates of ET-26 in different species. Data are presented as mean ± s.d., $n = 3$.

| measured concentration (ng ml$^{-1}$) | species | protein binding (%) |
|---|---|---|
| 654 | mouse | 95.47 ± 0.02* |
| | rat | 81.85 ± 1.00* |
| | dog | 65.53 ± 2.45 |
| | monkey | 61.33 ± 3.52 |
| | human | 62.82 ± 2.03 |
| $2.62 \times 10^3$ | mouse | 94.61 ± 0.06* |
| | rat | 85.03 ± 0.41* |
| | dog | 63.13 ± 5.52 |
| | monkey | 62.48 ± 4.48 |
| | human | 68.18 ± 1.78 |
| $5.23 \times 10^3$ | mouse | 92.98 ± 0.63* |
| | rat | 82.21 ± 1.47* |
| | dog | 69.40 ± 0.75 |
| | monkey | 64.34 ± 3.36 |
| | human | 73.92 ± 2.60 |

*$p < 0.05$, compared with PPB rates of human from the same measured concentration, using *post hoc* LSD *t*-test.

## 3.4. Tissue distribution in Sprague Dawley rats

In rats, both ET-26-HCl and ET-26-acid widely and rapidly distributed to several tissues (figure 6). CL of ET-26-HCl and ET-26-acid from tissues took around 3 h indicating that ET-26-HCl was a short-acting anaesthetic drug, consistent with the trend in plasma concentrations. Adipose tissue and liver had the highest concentration of ET-26-HCl followed by kidney, stomach, plasma, colon, spleen, intestines,

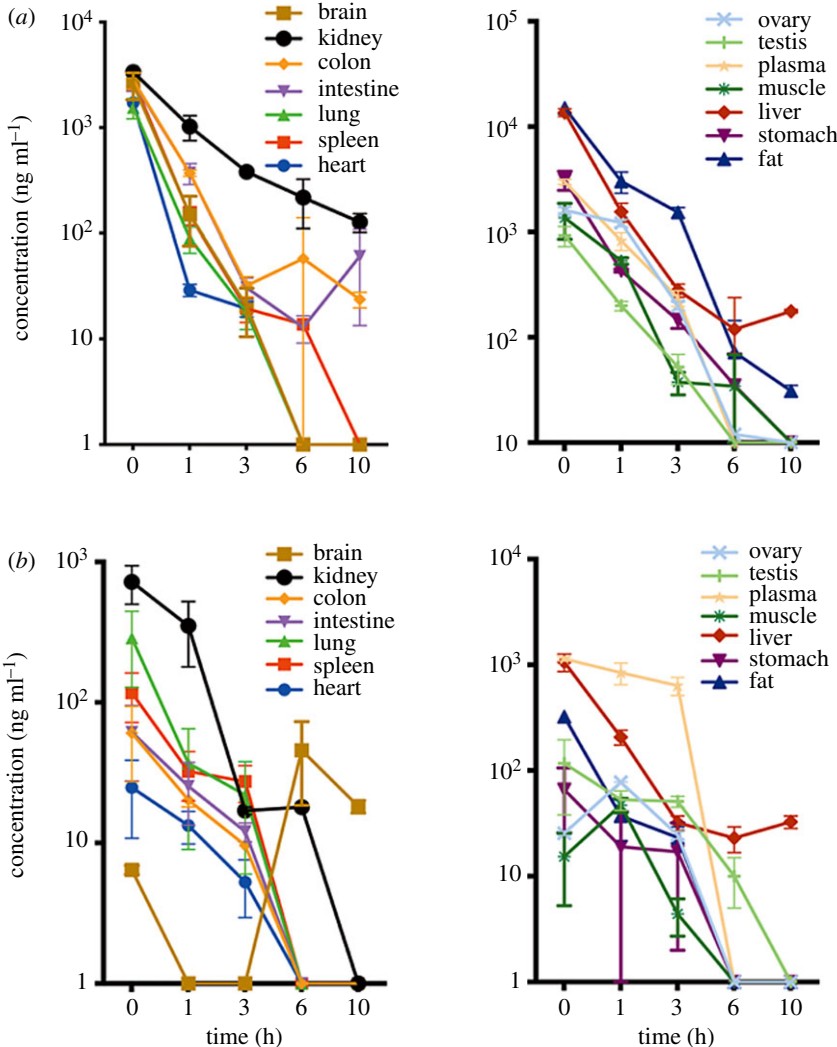

**Figure 6.** Concentration–time profile of (a) ET-26-HCl and (b) ET-26-acid in tissues following i.v. injection of 4.2 mg kg$^{-1}$ ET-26-HCl to rats (means ± s.d., $n = 6$).

**Table 3.** Main pharmacokinetic parameters of ET-26-HCl after intravenous administration of ET-26-HCl to beagle dogs. Data are presented as mean ± s.d., $n = 4$ for dogs. AUC$_{0-t}$, area under the plasma concentration–time curves from zero to the last measurable point; CL, clearance; C$_{max}$, maximum concentration; $t_{1/2}$, half-life; $V_d$, volume of distribution; $p$, $p$-value.

| parameters | dose (mg kg$^{-1}$) | | | $p$ |
| --- | --- | --- | --- | --- |
| | 1.045 | 2.09 | 4.18 | |
| dogs | | | | |
| $\lambda$ (1 h$^{-1}$) | 0.78 ± 0.06 | 0.54 ± 0.06 | 0.36 ± 0.06 | <0.01 |
| $t_{1/2}$ (h) | 0.92 ± 0.08 | 1.30 ± 0.18 | 2.17 ± 0.50 | <0.01 |
| C$_{max}$ (mg l$^{-1}$) | 1.48 ± 0.14 | 2.85 ± 0.22 | 6.23 ± 0.24 | <0.01 |
| AUC$_{0-t}$ (mg l$^{-1}$*min) | 1804.20 ± 118.26 | 3762.77 ± 108.50 | 7827.91 ± 447.44 | <0.01 |
| $V_d$ (l kg$^{-1}$) | 2.54 ± 0.29 | 3.40 ± 0.43 | 4.63 ± 0.67 | <0.01 |
| CL (l h$^{-1}$ kg$^{-1}$) | 1.92 ± 0.12 | 1.80 ± 0.06 | 1.50 ± 0.18 | <0.01 |

brain, heart, ovary, lung, muscle and testis. Meanwhile, the highest concentration of ET-26-acid was found in plasma and liver, followed by kidney, fat, lung, spleen, testis, stomach, intestines, colon, ovary, heart, muscle and brain.

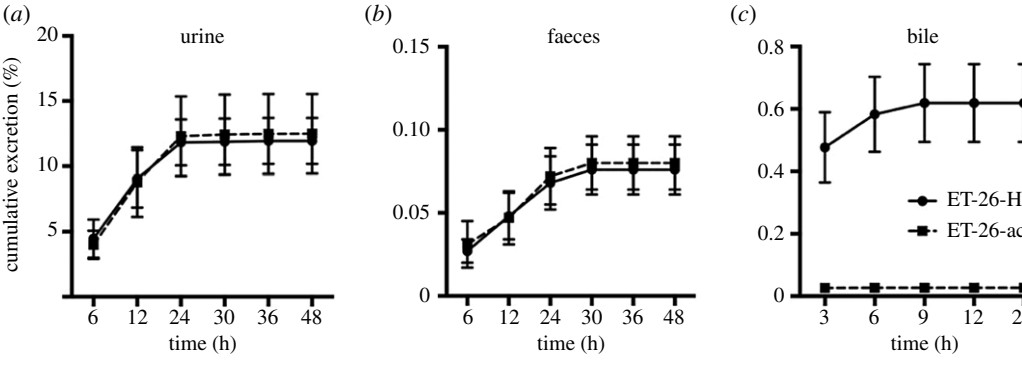

**Figure 7.** The (a) urinary (b) faecal and (c) biliary cumulative excretion of ET-26-HCl and ET-26-acid in rats (means ± s.d., $n = 8$) following i.vi administration at a dose of 4.2 mg kg$^{-1}$.

## 3.5. Excretion in Sprague Dawley rats

The accumulative excretion results of rats injected intravenously with ET-26-HCl (4.2 mg kg$^{-1}$) demonstrated that only a small fraction of ET-26-HCl and its main metabolite ET-26-acid were excreted through urine (ET-26-HCl, 11.97%; ET-26-acid, 12.49%), faeces (ET-26-HCl, 0.076%; ET-26-acid, 0.080%) and bile (ET-26-HCl, 0.619%; ET-26-acid, 0.027%). The 48 h of accumulative excretion of ET-26-HCl and ET-26-acid accounted for nearly 25% of the total dose (figure 7).

## 3.6. Discussion

ET-26-HCl, a promising short-acting anaesthetic agent, was investigated with regard to its distribution, metabolism, excretion and PK in animals in this study according to Product Development Under the Animal Rule: Guidance for Industry from FDA [16]. Several CYP-mediated metabolites were identified. Metabolically, ET-26-HCl was significantly more stable in human liver microsomes compared with microsomes from other species. A significant amount of parent compound was found in the human microsome, whereas there was none or little residual amount of ET-26-HCl in liver microsomes from other species, which were mainly *O*-demethylated and dehydrogenated *O*-demethylated metabolites. No human-specific metabolite was identified in this study. Relevant information on ET-26-HCl and its metabolites are summarized in table 4. Because of space limitation, only the most likely fragment ions are shown. The presumed metabolic pathways of ET-26-HCl in liver microsomes of all species are shown in figure 8. Based on the results of hepatic microsomal metabolic stability, possible metabolic pathway and metabolites, only dogs and rats were selected for PK analysis.

In our previous study, we had successfully established an LC–MS/MS method for quantification of ET-26-HCl and ET-26-acid in rat plasma and used it for a pharmacokinetic study [11]. In the present study, this simple, sensitive and reliable method was successfully used to evaluate metabolic stability, PK in beagle dogs, tissue distribution and excretion in SD rats, and PPB of ET-26-HCl. We observed rapid metabolism of ET-26-HCl with a half-life of less than 5 min in the liver microsomes from monkey, rat, mouse and dog. However, it was relatively stable with a half-life of 42.0 min in human liver microsomes. In beagles, the $t_{1/2}$ was about 1 h and $V_d$ was about 4 l kg$^{-1}$ (2.09 mg kg$^{-1}$), which suggested short residence time and rapid clearance of ET-26-HCl from the body. It is worth mentioning that there is a species difference for ET-26-HCl pharmacokinetics, since a large gap was observed in PK parameters, i.e. $t_{1/2}$, CL and $V_d$ of ET-26-HCl between dogs and rats (figure 5 and table 3; electronic supplementary material, figure S2 and table S4) [11]. ET-26-acid was almost undetectable in the plasma of dogs, most likely due to the difference in the type and expression of carboxylesterases between dogs and rats [11,17]. A greater than dose proportional increase in AUC and deceased clearance at higher doses were observed, which indicated the possible nonlinear PK of ET-26-HCl in dogs. Because hepatic clearance is the major pathway of elimination, the saturation of metabolism is considered to be the main reason.

The extrapolated hepatic clearances (CL$_{\text{hep, } in\ vivo}$ in table 1) are 19.1 and 296 ml min$^{-1}$ for rat and dog, respectively, which can be normalized to while the observed total body clearances are 1.78 and

**Table 4.** Identification of ET-26-HCl metabolites *in vitro* using UPLC-Q-TOF mass spectrometry.

| | description | retention time (min) | formula | formula change | observed mass (m/z) | main fragment ions | source (liver microsomes) |
|---|---|---|---|---|---|---|---|
| M0 | ET-26-HCl | 5.97 | $C_{15}H_{18}N_2O_3$ | null | 275.1395 | 171, 139, 113, 95 | human, dog |
| M1 | O-demethylation and desaturation | 5.41 | $C_{14}H_{14}N_2O_3$ | $-CH_4$ | 259.1075 | 157, 139, 95 | human, monkey, dog, rat, mouse |
| M2 | O-demethylation | 5.25 | $C_{14}H_{16}N_2O_3$ | $-CH_2$ | 261.1230 | 157, 139, 95 | human, monkey, dog, rat, mouse |
| M3 | desaturation | 6.12 | $C_{15}H_{16}N_2O_3$ | $-H_2$ | 273.1240 | 171, 113, 95 | human, dog |
| M4-1/M4-2 | demethylation, desaturation and hydroxylation/ O-demethylation, desaturation and hydroxylation | 5.14/5.36 | $C_{14}H_{14}N_2O_4$ | $-CH_4$, $+O$ | 275.1039 | 171, 113, 95, /231, 95 | monkey, dog, rat, mouse |
| M5 | O-demethylation and hydroxylation | 5.00 | $C_{14}H_{16}N_2O_4$ | $-CH_2$, $+O$ | 277.1178 | 173, 155, 95 | monkey, dog, rat, mouse |
| M6-1/M6-2 | hydroxylation and desaturation | 5.32/6.06 | $C_{15}H_{16}N_2O_4$ | $-H_2$, $+O$ | 289.1272 | 171, 113, 95/ 231, 187, 95 | dog, rat, mouse |

**Figure 8.** The presumed metabolic pathways of ET-26-HCl in liver microsomes of mouse, rat, beagle dog, monkey and human. H, human; Mk, monkey; D, dog; R, rat; Ms, mouse.

4.58 l h kg$^{-1}$ (assuming the average body weights are 0.25 kg and 10 kg for rats and dogs, respectively). The values are close to the observed clearances (table 3; electronic supplementary material, table S4). The results indicate that the prediction is reasonable and hepatic clearance is the major pathway for drug elimination.

As per Kleiber's law for allometric scaling between species, the animal metabolic rate can be scaled to around 0.75 power of the animal's size [18]. The relations in terms of clearance can be examined using the equation of $CL = a \times BW^b$, where $a$ is the typical value of clearance, BW is the average body weight of various species and $b$ is the power which should close to 0.75. Log-linear regression of the predicted clearance versus the average BW of mouse, rat, dog, monkey and human resolved that $a = 40$, $b = 0.74$ and $R^2 = 0.999$. Our results suggested a good fit using Kleiber's equation for allometric scaling (electronic supplementary material, figure S3).

In rats, ET-26-HCl was rapidly metabolized and distributed to blood-rich tissues and organs. The overall high concentrations of ET-26-HCl and ET-26-acid was observed in liver, kidney and plasma. However, the distribution into fat and brain was significantly higher for ET-26-HCl rather than ET-26-acid. Because ET-26-acid is more hydrophilic than ET-26-HCl, which may result in reduced passive permeability across cell membrane. Considerable amounts of ET-26-HCl in the brain (concentration $2.57 \times 10^3$ ng ml$^{-1}$ at the onset) suggest that it can effectively penetrate across BBB. The rapid elimination of ET-26-HCl from the brain within 3 h and complete excretion from kidney within 24 h, supporting the use of ET-26-HCl as a short-acting anaesthetic with rapid clearance. We believe that ET-26-HCl with its minimal adrenal suppression and potent sedative action in preclinical studies could be a promising short-acting anaesthetic drug candidate [6,7].

PPB of beagle dog, monkey and human were much lower than that of mouse and rat (table 2) indicating the inter-species variation between rodent and non-rodent. As a weak basic compound, albumin could be the major binding protein in the blood, similar to etomidate [19,20]. However, the exact binding sites need to be further identified. The pharmacokinetics of drugs with strong PPB (e.g.

phenytoin and warfarin) may be influenced by co-administration of ET-26-HCl due to its moderate binding rate in human plasma (68%) [21,22].

Less than 30% of ET-26-HCl was excreted through urine and faeces as parent drug or ET-26-acid, indicating that other important metabolic pathways should be considered. However, as the reference standards of proposed metabolites were not available, the proportion of each metabolite in the urine and faeces remains undetermined. Also, measurement of the kinetics of metabolite formation was not possible at this stage. Besides, pharmacological activity of the metabolites and their relative contribution to the total efficacy of ET-26-HCl warrant further clarification.

# 4. Conclusion

In this study, simple and sensitive UPLC-Q-OF-MS and LC–MS/MS methods were developed and successfully used to examine metabolic stability, metabolic products, PK, tissue distribution, excretion and PPB of ET-26-HCl, a novel etomidate analogue. We identified ET-26-HCl as a promising short-acting anaesthetic drug candidate based on its PK and distribution properties.

Ethics. All experiments were performed according to the Guidelines for the Care and Use of Laboratory Animals and approved by the Committee of Scientific Research and the Committee of Animal Care of the West China Hospital, Sichuan University (protocol no. 2015012311). Liver microsomes from human, monkey, dog, rat and mouse were purchased from Becton, Dickinson and Company. We were not required to complete an ethical assessment prior to conducting the researches involving liver microsomes. Human plasma was provided by Chengdu Fan Microanalysis Pharmaceutical Technology Co., Ltd and approved by the Ethical Committee of The Phase I Clinical Trial Research Office of the 416 Hospital of the Nuclear Industry, China (protocol no. LSD2018001).

Data accessibility. Data are available from the cited references and the electronic supplementary material.

Authors' contributions. L.Y., X.C., W.S.Z., W.W.X. and M.Y.X. prepared all samples for analysis. L.Y., X.C. and L.Z. collected and analysed the data. X.H.J and L.W. designed the study. L.Y. interpreted the results and wrote the manuscript. All authors gave final approval for publication and agreed to be accountable for all aspects of the work in ensuring that questions related to the accuracy or integrity of any part of the work are appropriately investigated and resolved.

Competing interests. We declare we have no competing interests.

Funding. We received no funding for this study.

Acknowledgements. The authors are indebted to the Neuroscience Research Center Anesthesia and Critical Emergency Research office at West China Hospital of Sichuan University, China for providing ET-26-HCl, and the Department of Clinical Pharmacy and Pharmacy Administration, Key Laboratory of Drug Ministry of Education, West China School of Pharmacy, Sichuan University, China for providing necessary facilities for carrying out this research work.

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
