## [Reviewer comments · Royal Society Open Science]

Review History

RSOS-191666.R0 (Original submission)

Review form: Reviewer 1

Is the manuscript scientifically sound in its present form?

Yes

Are the interpretations and conclusions justified by the results?

Yes

Is the language acceptable?

Yes

Do you have any ethical concerns with this paper?

No

Have you any concerns about statistical analyses in this paper?

No

Recommendation?

Accept with minor revision (please list in comments)

Comments to the Author(s)

This study is to examine the metabolite identification, tissue distribution, excretion and preclinical pharmacokinetic (PK) studies of ET-26-HCl. As a new analog of Etomidate, it may be used as a short acting anesthetic drug candidate with improved properties such as lower adrenal suppression, outstanding anesthetic effect and superior pharmacokinetic characteristics. In this study, the well developed and sensitive UPLC-Q-OF-MS and LC-MS/MS were successfully used to detect metabolic stability, metabolic products, PK, tissue distribution, excretion and PPB of ET-26-HCl. Overall, the results including in vitro and in vivo preclinical datasets are well presented and the manuscript is well written. Some minor comments below to improve the manuscript.

Comments:

1. Page 14, the "PK in Beagle Dogs" results section: Regardless of doses, once ET-26-HCl is administered, the systemic PK reflects that of the same molecular entity. Therefore, there are differences in the estimated λ , Vd and CL term for three doses, are that due to possible non-linearity of the drug? Same question for Table 3.

Minor:

1. Page 7, line 107. It should have space between "=" and "0.693 x t1/2 / Cprotein".

2. For the PK study of beagle dogs and excretion test of rats, how do you identify those doses? As you described, beagle dogs in the 6 groups received three doses of ET-26-HCl: 1.045 mg/kg, 2.090 mg/kg, and 4.180 mg/kg as low (L), medium (M), and high (H) dose; however, the rats received a single 4.2 mg/kg dose of ET-26-HCl. Please provide the citations or calculation of those doses.

3. Page 11 line 197 and Fig 2. For the "metabolic stability" result, what's the baseline of the ET-26-HCl metabolic stability? You may need to include the control group which without liver microsomes.

4. Figure 5 and 6: All PK data should be shown on a log-y axis.

5. Page 18, line 326 - 327: Please add the citation or evidence for the description "As a weak basic compound, albumin could be the major binding protein in the blood, similar to etomidate".

6. Page 19: Last sentence of your conclusion, you mentioned the compound possesses lower adrenal suppression. Please add some evidence or citations in the discussion section.

Review form: Reviewer 2

Is the manuscript scientifically sound in its present form?

No

Are the interpretations and conclusions justified by the results?

Yes

Is the language acceptable?

Yes

Do you have any ethical concerns with this paper?

No

Have you any concerns about statistical analyses in this paper?

Yes

Recommendation?

Major revision is needed (please make suggestions in comments)

Comments to the Author(s)

This paper reports the preclinical pharmacokinetics of ET-26-HCl. However, some points should be clarified or revised before considering its publication. My points are detailed below.

1. Line 107; Equation 1 should be revised to " $= 0.693/t_{1/2}/C_{\text{protein}}$ ".
2. Line 108; The "CL" could be more specified. i.e. "CL_{int, in vivo}". The term should be revised thereafter throughout the manuscript.
3. Line 112; The "fu" should be unbound fraction of the drug in blood. I cannot agree with the sentence "fu= 1 is generally used". The meaning of this sentence is unclear. Do you mean you consider fu value of all the species is 1? It is not true. Based on plasma protein binding value for each species (Table 2), fu values are much smaller than 1 and vary depend on species. Therefore, fu value applied for each species should be specified.
4. Line 192; The unit for CL_{int, in vitro} values listed here is different from that on Table 1.
5. Line 258-259; This sentence should be revised based on statistical analysis of data in Table 3. I cannot agree that CL values are dose-dependent. They seem to be dose-independent. Also, The V_d values in rats seem to be constant over the dose ranges applied.
6. Line 271-272; The recovered amount listed as percentage of dose seem to be values for ET-26-HCl and ET-26-acid (metabolite). However, the values in parenthesis listed values for ET-26-HCl twice. One should be revised as "ET-26-acid"?
7. Line 298-300; Please refer my comments 5 and revise this sentence.
8. Fig. 2; I recommend using log scale for y-axis to show the half-life. What is the meaning of "(ng/mL)"?
9. Table 1; Symbol for clearances should be revised. Subscript was misused. The unit for CL_{int, in vitro} values listed here is different from the text (Refer my comment 4). I recommend using "Extrapolated CL_{int, in vivo}" instead of "Extrapolated CL_{int, in vitro}"
10. Table 3; Data should be shown with statistical analysis results to verify the dose-dependency of pharmacokinetics of ET-26-HCl.

Decision letter (RSOS-191666.R0)

19-Nov-2019

Dear Professor Wang:

Title: Metabolite Identification, Tissue Distribution, Excretion and Preclinical Pharmacokinetic Studies of ET-26-HCl, a New Analog of Etomidate
 Manuscript ID: RSOS-191666

The editor assigned to your manuscript has now received comments from reviewers. We would like you to revise your paper in accordance with the referee and Subject Editor suggestions which can be found below (not including confidential reports to the Editor). Please note this decision does not guarantee eventual acceptance.

Please submit your revised paper before 12-Dec-2019. Please note that the revision deadline will

expire at 00.00am on this date. If we do not hear from you within this time then it will be assumed that the paper has been withdrawn. In exceptional circumstances, extensions may be possible if agreed with the Editorial Office in advance. We do not allow multiple rounds of revision so we urge you to make every effort to fully address all of the comments at this stage. If deemed necessary by the Editors, your manuscript will be sent back to one or more of the original reviewers for assessment. If the original reviewers are not available we may invite new reviewers.

RSC Associate Editor:
Comments to the Author:
(There are no comments.)

RSC Subject Editor:
Comments to the Author:
(There are no comments.)

Reviewers' Comments to Author:
Reviewer: 1

Comments to the Author(s)

This study is to examine the metabolite identification, tissue distribution, excretion and preclinical pharmacokinetic (PK) studies of ET-26-HCl. As a new analog of Etomidate, it may be used as a short acting anesthetic drug candidate with improved properties such as lower adrenal suppression, outstanding anesthetic effect and superior pharmacokinetic characteristics. In this

study, the well developed and sensitive UPLC-Q-OF-MS and LC-MS/MS were successfully used to detect metabolic stability, metabolic products, PK, tissue distribution, excretion and PPB of ET-26-HCl. Overall, the results including in vitro and in vivo preclinical datasets are well presented and the manuscript is well written. Some minor comments below to improve the manuscript.

Comments:

1. Page 14, the “PK in Beagle Dogs” results section: Regardless of doses, once ET-26-HCl is administered, the systemic PK reflects that of the same molecular entity. Therefore, there are differences in the estimated λ , V_d and CL term for three doses, are that due to possible non-linearity of the drug? Same question for Table 3.

Minor:

1. Page 7, line 107. It should have space between “=” and “ $0.693 \times t_{1/2} / C_{\text{protein}}$ ”.

2. For the PK study of beagle dogs and excretion test of rats, how do you identify those doses? As you described, beagle dogs in the 6 groups received three doses of ET-26-HCl: 1.045 mg/kg, 2.090 mg/kg, and 4.180 mg/kg as low (L), medium (M), and high (H) dose; however, the rats received a single 4.2 mg/kg dose of ET-26-HCl. Please provide the citations or calculation of those doses.

3. Page 11 line 197 and Fig 2. For the “metabolic stability” result, what’s the baseline of the ET-26-HCl metabolic stability? You may need to include the control group which without liver microsomes.

4. Figure 5 and 6: All PK data should be shown on a log-y axis.

5. Page 18, line 326 – 327: Please add the citation or evidence for the description “As a weak basic compound, albumin could be the major binding protein in the blood, similar to etomidate”.

6. Page 19: Last sentence of your conclusion, you mentioned the compound possesses lower adrenal suppression. Please add some evidence or citations in the discussion section.

Reviewer: 2

Comments to the Author(s)

This paper reports the preclinical pharmacokinetics of ET-26-HCl. However, some points should be clarified or revised before considering its publication. My points are detailed below.

1. Line 107; Equation 1 should be revised to “ $= 0.693/t_{1/2}/C_{\text{protein}}$ ”.

2. Line 108; The “CL” could be more specified. i.e. “CL_{int, in vivo}”. The term should be revised thereafter throughout the manuscript.

3. Line 112; The “ f_u ” should be unbound fraction of the drug in blood. I cannot agree with the sentence “ $f_u = 1$ is generally used”. The meaning of this sentence is unclear. Do you mean you consider f_u value of all the species is 1? It is not true. Based on plasma protein binding value for each species (Table 2), f_u values are much smaller than 1 and vary depend on species. Therefore, f_u value applied for each species should be specified.

4. Line 192; The unit for CL_{int, in vitro} values listed here is different from that on Table 1.

5. Line 258-259; This sentence should be revised based on statistical analysis of data in Table 3. I cannot agree that CL values are dose-dependent. They seem to be dose-independent. Also, The V_d values in rats seem to be constant over the dose ranges applied.

6. Line 271-272; The recovered amount listed as percentage of dose seem to be values for ET-26-HCl and ET-26-acid (metabolite). However, the values in parenthesis listed values for ET-26-HCl twice. One should be revised as “ET-26-acid”?

7. Line 298-300; Please refer my comments 5 and revise this sentence.

8. Fig. 2; I recommend using log scale for y-axis to show the half-life. What is the meaning of “(ng/mL)”?

9. Table 1; Symbol for clearances should be revised. Subscript was misused. The unit for CL_{int, in vitro} values listed here is different from the text (Refer my comment 4). I recommend using “Extrapolated CL_{int, in vivo}” instead of “Extrapolated CL_{int, in vitro}”

10. Table 3; Data should be shown with statistical analysis results to verify the dose-dependency of pharmacokinetics of ET-26-HCl.

Author's Response to Decision Letter for (RSOS-191666.R0)

See Appendix A.

Decision letter (RSOS-191666.R1)

17-Dec-2019

Dear Professor Wang:

Title: Metabolite Identification, Tissue Distribution, Excretion and Preclinical Pharmacokinetic Studies of ET-26-HCl, a New Analog of Etomidate
Manuscript ID: RSOS-191666.R1

It is a pleasure to accept your manuscript in its current form for publication in Royal Society Open Science. The chemistry content of Royal Society Open Science is published in collaboration with the Royal Society of Chemistry.

RSC Associate Editor
Comments to the Author:
(There are no comments.)

Reviewer(s)' Comments to Author:

Appendix A

Responses to the Reviewers:

First, we would like to thank the reviewers for their constructive comments. We have carefully addressed the reviewers' concerns and made corresponding revisions. The changes to our manuscript within the document were highlighted by using red colored text. Detailed descriptions related to the revisions are given as follows:

Reviewer 1's Comments, Point 1:

Page 14, the "PK in Beagle Dogs" results section: Regardless of doses, once ET-26- HCl is administrated, the systemic PK reflects that of the same molecular entity. Therefore, there are differences in the estimated lambda (λ), Vd and CL term for three doses, are that due to possible non-linearity of the drug? Same question for Table 3.

Response:

Thanks for the suggestion. Since the AUC increased more than dose proportionality and clearance decreased at higher doses, the nonlinearity is very likely caused by the saturation of drug metabolism which is the major elimination pathway of ET-26-HCl. We have included some statements regarding this issue (see Discussion). (Page 15, line 298-301)

Reviewer 1's Comments, Point 2:

Page 7, line 107. It should have space between "=" and "0.693 x t1/2 / Cprotein".

Response:

Thanks. We have changed the formula in the correct way according to the reference [1]. (Page 6, line 106)

Reference

[1] Zhang C, Zhang X, Wang G, et al. 2018 Preclinical Pharmacokinetics of C118P, a Novel Prodrug of Microtubules Inhibitor and Its Metabolite C118 in Mice, Rats, and Dogs. *Molecules*, **23**(11): 2883.

Reviewer 1's Comments, Point 3:

For the PK study of beagle dogs and excretion test of rats, how do you identify those doses? As you described, beagle dogs in the 6 groups received three doses of ET-26- HCl: 1.045 mg/kg, 2.090 mg/kg, and 4.180 mg/kg as low (L), medium (M), and high (H) dose; however, the rats received a single 4.2 mg/kg dose of ET-26-HCl. Please provide the citations or calculation of those doses.

Response:

Thank the reviewer for this comment. According to the “Guidelines for non-clinical pharmacokinetic studies of drugs” of China Food and Drug Administration, the in vivo pharmacokinetic studies of animals should be performed with at least three dose groups, which the high dose is preferably close to the maximum tolerated dose, and the small dose is selected based on the limits of the effective dose of the animals. The ED50 of ET-26-HCl was 2.1 mg/kg and 1.045 mg/kg for rat and dog, respectively, according to the results of the drug effect in the early stage of the cooperation laboratory. In our pre-experiment, individual rats appeared myoclonus after the 4-fold ED50 dose administration. When given the higher dose, some rats had convulsions and even died. Thus, the ED50, 2-fold ED50, and 4-fold ED50 dose were used as the low, middle and high dose group for the pharmacokinetic study in our previous research [2]. Similar dosage design was used in our present work. The middle dose (4.2 mg/kg dose of ET-26-HCl) was utilized for the excretion test in rats.

Reference

[2] Chen X, Zhang W, Rios S, et al. 2017 An HPLC tandem mass spectrometry for quantification of ET-26-HCl and its major metabolite in plasma and application to a pharmacokinetic study in rats. *Journal of Pharmaceutical & Biomedical Analysis* **149**, 381.

Reviewer 1’s Comments, Point 4:

Page 11 line 197 and Fig 2. For the “metabolic stability” result, what’s the baseline of the ET-26-HCl metabolic stability? You may need to include the control group which without liver microsomes.

Response:

Thanks for your suggestion. We have added the blank group without liver microsomes as the control in Fig 2 of the revision. (Page 21)

Reviewer 1's Comments, Point 5:

Figure 5 and 6: All PK data should be shown on a log-y axis.

Response:

We have changed the y axis according to your suggestion. Thank you! To prevent misleading, we moved the concentration-time profiles of rats in Fig.5 which was cited from our previous study to Fig.S2 in the supplementary materials. (Page 24, 25).

Reviewer 1's Comments, Point 6:

Page 18, line 326 – 327: Please add the citation or evidence for the description “As a weak basic compound, albumin could be the major binding protein in the blood, similar to etomidate”.

Response:

Thanks for your suggestion. We have added the citations in the revision. (Page 17, line 327)

Reviewer 1's Comments, Point 7:

Page 19: Last sentence of your conclusion, you mentioned the compound possesses lower adrenal suppression. Please add some evidence or citations in the discussion section.

Response:

Thank you for your comment. We have rewritten the conclusion section in the revision. The evidence of “the compound possesses lower adrenal suppression” has been mentioned in the introduction section with some citations (Compared to etomidate, ET-26-HCl exhibited

superior anesthetic property, reduced adrenal suppression, and optimal myocardial performance [6-10]). (Page 3, line 50-51; Page 17-18, line 341-343)

Reviewer 2's Comments, Point 1:

Line 107; Equation 1 should be revised to “ = 0.693/t_{1/2}/C_{protein}”.

Response:

Thanks. We have changed the formula in the correct way according to your comment. (Page 6, line 106)

Reviewer 2's Comments, Point 2:

Line 108; The “CL” could be more specified. i.e. “CL_{int, in vivo}”. The term should be revised thereafter throughout the manuscript.

Response:

According to your suggestion, we have corrected the “CL” into “CL_{int, in vivo}” throughout the manuscript. Thanks! (Page 6, line 107)

Reviewer 2's Comments, Point 3:

Line 112; The “f_u” should be unbound fraction of the drug in blood. I cannot agree with the sentence “f_u= 1 is generally used”. The meaning of this sentence is unclear. Do you mean you consider f_u value of all the species is 1? It is not true. Based on plasma protein binding value for each species (Table 2), f_u values are much smaller than 1 and vary depend on species. Therefore, f_u value applied for each species should be specified.

Response:

Thanks for your careful checks. It was a typo and we have made the correction in the new version. (Page 6, line 111)

Reviewer 2's Comments, Point 4:

Line 192; The unit for CL_{int}, in vitro values listed here is different from that on Table 1.

Response:

Thanks! We have corrected these mistakes. (Page 10, line 192)

Reviewer 2's Comments, Point 5:

Line 258-259; This sentence should be revised based on statistical analysis of data in Table 3. I cannot agree that CL values are dose-dependent. They seem to be dose-independent. Also, The V_d values in rats seem to be constant over the dose ranges applied.

Response:

Our statement about the dose-dependent clearance is misleading, which actually should mean that the clearance decreases in a dose-dependent manner. To avoid misunderstanding, we changed the statement as “clearance (CL) decreased at higher doses” in the revised version. (Page 13, line 256-257)

Reviewer 2's Comments, Point 6:

Line 271-272; The recovered amount listed as percentage of dose seem to be values for ET-26-HCl and ET-26-acid (metabolite). However, the values in parenthesis listed values for ET-26-HCl twice. One should be revised as “ET-26-acid”?

Response:

Thanks. We have corrected these mistakes based on your suggestion. (Page 14, line 269-270)

Reviewer 2's Comments, Point 7:

Line 298-300; Please refer my comments 5 and revise this sentence.

Response:

Thanks. We have revised this sentence according to your suggestion. (Page 15, line 298-301)

Reviewer 2's Comments, Point 8:

Fig. 2; I recommend using log scale for y-axis to show the half-life. What is the meaning of “(ng/mL)”?

Response:

We were sorry for our careless mistake of “(ng/mL)”. We have changed the y-axis to the log scale accordingly in the revised Fig. 2. (Page 21, Fig. 2)

Reviewer 2's Comments, Point 9:

Table 1; Symbol for clearances should be revised. Subscript was misused. The unit for CL_{int}, in vitro values listed here is different from the text (Refer my comment 4). I recommend using “Extrapolated CL_{int}, in vivo” instead of “Extrapolated CL_{int}, in vitro”

Response:

Thanks for your correction. We have corrected it according to your suggestion. (Page 28, Table 1)

Reviewer 2's Comments, Point 10:

Table 3; Data should be shown with statistical analysis results to verify the dose-dependency of pharmacokinetics of ET-26-HCl.

Response:

Thank you for your suggestion. We have added the p value of each pharmacokinetics parameters in the revised Table 3. To prevent misleading, we moved the data of rats cited from our previous study to Table S4 in the supplementary materials. (Page 30, Table 3)